# Softmax Deep Double Deterministic Policy Gradients

**Ling Pan**[1], **Qingpeng Cai**[2], **Longbo Huang**[1]
[1]Institute for Interdisciplinary Information Sciences, Tsinghua University
pl17@mails.tsinghua.edu.cn, longbohuang@tsinghua.edu.cn
[2]Alibaba Group
qingpeng.cqp@alibaba-inc.com

## Abstract

A widely-used actor-critic reinforcement learning algorithm for continuous control, Deep Deterministic Policy Gradients (DDPG), suffers from the overestimation problem, which can negatively affect the performance. Although the state-of-the-art Twin Delayed Deep Deterministic Policy Gradient (TD3) algorithm mitigates the overestimation issue, it can lead to a large underestimation bias. In this paper, we propose to use the Boltzmann softmax operator for value function estimation in continuous control. We first theoretically analyze the softmax operator in continuous action space. Then, we uncover an important property of the softmax operator in actor-critic algorithms, i.e., it helps to smooth the optimization landscape, which sheds new light on the benefits of the operator. We also design two new algorithms, Softmax Deep Deterministic Policy Gradients (SD2) and Softmax Deep Double Deterministic Policy Gradients (SD3), by building the softmax operator upon single and double estimators, which can effectively improve the overestimation and underestimation bias. We conduct extensive experiments on challenging continuous control tasks, and results show that SD3 outperforms state-of-the-art methods.

## 1 Introduction

Deep Deterministic Policy Gradients (DDPG) [24] is a widely-used reinforcement learning [26, 30, 24, 29] algorithm for continuous control, which learns a deterministic policy using the actor-critic method. In DDPG, the parameterized actor network learns to determine the best action with highest value estimates according to the critic network by policy gradient descent. However, as shown recently in [15], one of the dominant concerns for DDPG is that it suffers from the overestimation problem as in the value-based Q-learning [37] method, which can negatively affect the performance with function approximation [34]. Therefore, it is of vital importance to have good value estimates, as a better estimation of the value function for the critic can drive the actor to learn a better policy.

To address the problem of overestimation in actor-critic, Fujimoto et al. propose the Twin Delayed Deep Deterministic Policy Gradient (TD3) method [15] leveraging double estimators [20] for the critic. However, directly applying the Double Q-learning [20] algorithm, though being a promising method for avoiding overestimation in value-based approaches, cannot fully alleviate the problem in actor-critic methods. A key component in TD3 [15] is the Clipped Double Q-learning algorithm, which takes the minimum of two Q-networks for value estimation. In this way, TD3 significantly improves the performance of DDPG by reducing the overestimation. Nevertheless, TD3 can lead to a large underestimation bias, which also impacts performance [10].

The Boltzmann softmax distribution has been widely adopted in reinforcement learning. The softmax function can be used as a simple but effective action selection strategy, i.e., Boltzmann exploration [33, 9], to trade-off exploration and exploitation. In fact, the optimal policy in entropy-regularized reinforcement learning [18, 19] is also in the form of softmax. Although it has been long believed that the softmax operator is not a non-expansion and can be problematic when used to update value

functions [25, 4], a recent work [32] shows that the difference between the value function induced by the softmax operator and the optimal one can be controlled in discrete action space. In [32], Song et al. also successfully apply the operator for value estimation in deep Q-networks [26], and show the promise of the operator in reducing overestimation. However, the proof technique in [32] does not always hold in the continuous action setting and is limited to discrete action space. Therefore, there still remains a theoretical challenge of whether the error bound can be controlled in the continuous action case. In addition, we find that the softmax operator can also be beneficial when there is no overestimation bias and with enough exploration noise, while previous works fail to understand the effectiveness of the operator in such cases.

In this paper, we investigate the use of the softmax operator in updating value functions in actor-critic methods for continuous control, and show that it has several advantages that makes it appealing. Firstly, we theoretically analyze the properties of the softmax operator in continuous action space. We provide a new analysis showing that the error between the value function under the softmax operator and the optimal can be bounded. The result paves the way for the use of the softmax operator in deep reinforcement learning with continuous action space, despite that previous works have shown theoretical disadvantage of the operator [25, 4]. Then, we propose to incorporate the softmax operator into actor-critic for continuous control. We uncover a fundamental impact of the softmax operator, i.e., it can smooth the optimization landscape and thus helps learning empirically. Our finding sheds new light on the benefits of the operator, and properly justifies its use in continuous control.

We first build the softmax operator upon single estimator, and develop the Softmax Deep Deterministic Policy Gradient (SD2) algorithm. We demonstrate that SD2 can effectively reduce overestimation and outperforms DDPG. Next, we investigate the benefits of the softmax operator in the case where there is underestimation bias on top of double estimators. It is worth noting that a direct combination of the operator with TD3 is ineffective and can only worsen the underestimation bias. Based on a novel use of the softmax operator, we propose the Softmax Deep Double Deterministic Policy Gradient (SD3) algorithm. We show that SD3 leads to a better value estimation than the state-of-the-art TD3 algorithm, where it can improve the underestimation bias, and results in better performance and higher sample efficiency.

We conduct extensive experiments in standard continuous control tasks from OpenAI Gym [6] to evaluate the SD3 algorithm. Results show that SD3 outperforms state-of-the-art methods including TD3 and Soft Actor-Critic (SAC) [19] with minimal additional computation cost.

## 2 Preliminaries

The reinforcement learning problem can be formulated by a Markov decision process (MDP) defined as a 5-tuple $(\mathcal{S}, \mathcal{A}, r, p, \gamma)$, with $\mathcal{S}$ and $\mathcal{A}$ denoting the set of states and actions, $r$ the reward function, $p$ the transition probability, and $\gamma$ the discount factor. We consider a continuous action space, and assume it is bounded. We also assume the reward function $r$ is continuous and bounded, where the assumption is also required in [31]. In continuous action space, taking the $\max$ operator over $\mathcal{A}$ as in Q-learning [37] can be expensive. DDPG [24] extends Q-learning to continuous control based on the Deterministic Policy Gradient [31] algorithm, which learns a deterministic policy $\pi(s; \phi)$ parameterized by $\phi$ to maximize the Q-function to approximate the $\max$ operator. The objective is to maximize the expected long-term rewards $J(\pi(\cdot; \phi)) = \mathbb{E}[\sum_{k=0}^{\infty} \gamma^k r(s_k, a_k) | s_0, a_0, \pi(\cdot; \phi)]$. Specifically, DDPG updates the policy by the deterministic policy gradient, i.e.,

$$\nabla_\phi J(\pi(\cdot; \phi)) = \mathbb{E}_s \left[ \nabla_\phi(\pi(s; \phi)) \nabla_a Q(s, a; \theta)|_{a=\pi(s;\phi)} \right], \tag{1}$$

where $Q(s, a; \theta)$ is the Q-function parameterized by $\theta$ which approximates the true parameter $\theta^{\text{true}}$. We let $\mathcal{T}(s')$ denote the value estimation function, which is used to estimate the target Q-value $r + \gamma\mathcal{T}(s')$ for state $s'$. Then, we see that DDPG updates its critic according to $\theta' = \theta + \alpha\mathbb{E}_{s,a\sim\rho} (r + \gamma\mathcal{T}_{\text{DDPG}}(s') - Q(s, a; \theta)) \nabla_\theta Q(s, a; \theta)$, where $\mathcal{T}_{\text{DDPG}} = Q(s', \pi(s'; \phi^-); \theta^-)$, $\rho$ denotes the sample distribution from the replay buffer, $\alpha$ is the learning rate, and $\phi^-, \theta^-$ denote parameters of the target networks for the actor and critic respectively.

## 3 Analysis of the Softmax Operator in Continuous Action Space

In this section, we theoretically analyze the softmax operator in continuous action space by studying the performance bound of value iteration under the operator.

The softmax operator in continuous action space is defined by $\operatorname{softmax}_\beta(Q(s,\cdot)) = \int_{a\in\mathcal{A}}\frac{\exp(\beta Q(s,a))}{\int_{a'\in\mathcal{A}}\exp(\beta Q(s,a'))da'}Q(s,a)da$, where $\beta$ is the parameter of the softmax operator.

In Theorem 1, we provide an $O(1/\beta)$ upper bound for the difference between the max and softmax operators. The result is helpful for deriving the error bound in value iteration with the softmax operator in continuous action space. The proof of Theorem 1 is in Appendix A.1.

**Theorem 1** *Let* $\mathcal{C}(Q,s,\epsilon) = \{a|a\in\mathcal{A},Q(s,a)\ge\max_a Q(s,a)-\epsilon\}$ *and* $F(Q,s,\epsilon) = \int_{a\in\mathcal{C}(Q,s,\epsilon)}1da$ *for any* $\epsilon > 0$ *and any state* $s$. *The difference between the max operator and the softmax operator is* $0\le\max_a Q(s,a)-\operatorname{softmax}_\beta(Q(s,\cdot))\le\frac{\int_{a\in\mathcal{A}}1da-1-\ln F(Q,s,\epsilon)}{\beta}+\epsilon$.

**Remark.** A recent work [32] studies the distance between the two operators in the discrete setting. However, the proof technique in [32] is limited to discrete action space. This is because applying the technique requires that for any state $s$, the set of the maximum actions at $s$ with respect to the Q-function $Q(s,a)$ covers a continuous set, which often only holds in special cases in the setting with continuous action. In this case, $\epsilon$ can be 0 and the bound still holds, which turns into $\frac{\int_{a\in\mathcal{A}}1da-1-\ln F(Q,s,0)}{\beta}$. Note that when $Q(s,a)$ is a constant function with respect to $a$, the upper bound in Theorem 1 will be 0, where the detailed discussion is in Appendix A.1.

Now, we formally define value iteration with the softmax operator by $Q_{t+1}(s,a) = r_t(s,a) + \gamma\mathbb{E}_{s'\sim p(\cdot|s,a)}[V_t(s')]$, $V_{t+1}(s) = \operatorname{softmax}_\beta(Q_{t+1}(s,\cdot))$, which updates the value function using the softmax operator iteratively. In Theorem 2, we provide an error bound between the value function under the softmax operator and the optimal in continuous action space, where the proof can be found in Appendix A.2.

**Theorem 2** *For any iteration $t$, the difference between the optimal value function $V^*$ and the value function induced by softmax value iteration at the $t$-th iteration $V_t$ satisfies:*

$$||V_t-V^*||_\infty\le\gamma^t||V_0(s)-V^*(s)||_\infty+\frac{1}{1-\gamma}\frac{\beta\epsilon+\int_{a\in\mathcal{A}}1da-1}{\beta}-\sum_{k=1}^{t}\gamma^{t-k}\frac{\min_s\ln F(Q_k,s,\epsilon)}{\beta}.$$

Therefore, for any $\epsilon > 0$, the error between the value function induced by the softmax operator and the optimal can be bounded, which converges to $\epsilon/(1-\gamma)$, and can be arbitrarily close to 0 as $\beta$ approaches to infinity. Theorem 2 paves the way for the use of the softmax operator for value function updates in continuous action space, as the error can be controlled in a reasonable scale.

## 4  The Softmax Operator in Actor-Critic

In this section, we propose to employ the softmax operator for value function estimation in standard actor-critic algorithms with single estimator and double estimators. We first show that the softmax operator can smooth the optimization landscape and help learning empirically. Then, we show that it enables a better estimation of the value function, which effectively improves the overestimation and underestimation bias when built upon single and double estimators respectively.

### 4.1  The Softmax Operator Helps to Smooth the Optimization Landscape

We first show that the softmax operator can help to smooth the optimization landscape. For simplicity, we showcase the smoothing effect based on a comparative study of DDPG and our new SD2 algorithm (to be introduced in Section 4.2). SD2 is a variant of DDPG that leverages the softmax operator to update the value function, which is the only difference between the two algorithms. We emphasize that the smoothing effect is attributed to the softmax operator, and also holds for our proposed SD3 algorithm (to be introduced in Section 4.3), which uses the operator to estimate value functions.[1]

We design a toy 1-dimensional, continuous state and action environment MoveCar (Figure 1(a)) to illustrate the effect. The car always starts at position $x_0 = 8$, and can take actions ranging in $[-1.0, 1.0]$ to move left or right, where the left and right boundaries are 0 and 10. The rewards are

$+2$, $+1$ in neighboring regions centered at $x_1 = 1$ and $x_2 = 9$, respectively, with the length of the neighboring region to be 1. In other positions, the reward is 0. The episode length is 100 steps. We run DDPG and SD2 on MoveCar for 100 independent runs. To exclude the effect of the exploration, we add a gaussian noise with the standard deviation to be high enough to the action during training, and both two algorithms collects diverse samples in the warm-up phase where actions are sampled from a uniform distribution. More details about the experimental setup are in Appendix B.2. The performance result is shown in Figure 1(b), where the shaded area denotes half a standard deviation for readability. As shown, SD2 outperforms DDPG in final performance and sample efficiency.

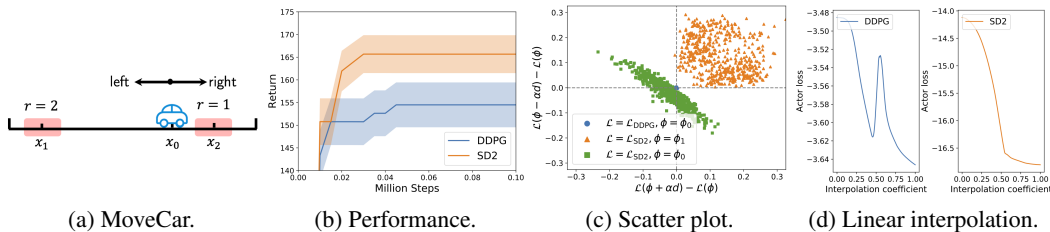

| (a) MoveCar. | (b) Performance. | (c) Scatter plot. | (d) Linear interpolation. |

Figure 1: Analysis of smoothing effect in the MoveCar environment.

We investigate the optimization landscape based on the visualization technique proposed in [2]. According to Eq. (1), we take the loss function of the actor $\mathcal{L}(\phi) = \mathbb{E}_{s \sim \rho} [-Q(s, \pi(s; \phi); \theta)]$ as the objective function in our analysis. To understand the local geometry of the actor losses $\mathcal{L}_{\text{DDPG}}$ and $\mathcal{L}_{\text{SD2}}$, we randomly perturb the corresponding policy parameters $\phi_0$ and $\phi_1$ learned by DDPG and SD2 during training from a same random initialization. Specifically, the key difference between the two parameters is that $\phi_0$ takes an action to move left in locations $[0, 0.5]$, while $\phi_1$ determines to move right. Thus, $\phi_1$ are better parameters than $\phi_0$. The random perturbation is obtained by randomly sampling a batch of directions $d$ from a unit ball, and then perturbing the policy parameters in positive and negative directions by $\phi \pm \alpha d$ for some value $\alpha$. Then, we evaluate the difference between the perturbed loss functions and the original loss function, i.e., $\mathcal{L}(\phi \pm \alpha d) - \mathcal{L}(\phi)$.

Figure 1(c) shows the scatter plot of random perturbation. For DDPG, the perturbations for its policy parameters $\phi_0$ are close to zero (blue circles around the origin). This implies that there is a flat region in $\mathcal{L}_{\text{DDPG}}$, which can be difficult for gradient-based methods to escape from [12]. Figure 2(a) shows that the policy of DDPG converges to always take action $-1$ at each location. On the other hand, as all perturbations around the policy parameters $\phi_1$ of SD2 with respect to its corresponding loss function $\mathcal{L}_{\text{SD2}}$ are positive (orange triangles), the point $\phi_1$ is a local minimum. Figure 2(b) confirms that SD2 succeeds to learn an optimal policy to move the agent to high-reward region $[0.5, 1.5]$. To illustrate the critical effect of the softmax operator on the objective, we also evaluate the change of the loss function $\mathcal{L}_{\text{SD2}}(\phi_0)$ of SD2 with respect to parameters $\phi_0$ from DDPG. Figure 1(c) shows that $\phi_0$ is in an almost linear region under $\mathcal{L}_{\text{SD2}}$ (green squares), and the loss can be reduced following several directions, which demonstrates the benefits of optimizing the softmax version of the objective.

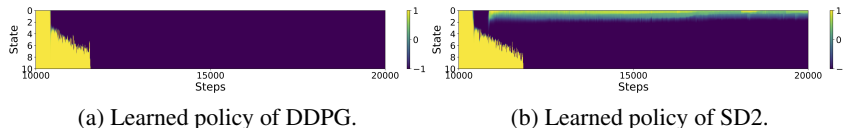

| (a) Learned policy of DDPG. | (b) Learned policy of SD2. |

Figure 2: Policies of DDPG and SD2 during learning for each state (y-axis) at each step (x-axis).

To further analyze the difficulty of the optimization of $\mathcal{L}_{\text{DDPG}}$ on a more global view, we linearly interpolate between the parameters of the policies from DDPG and SD2, i.e., $\alpha\phi_0 + (1 - \alpha)\phi_1$ ($0 \leq \alpha \leq 1$) as in [16, 2]. Figure 1(d) illustrates the result with varying values of $\alpha$. As shown, there exists at least a monotonically decreasing path in the actor loss of SD2 to a good solution. As a result, the smoothing effect of the softmax operator on the optimization landscape can help learning and reduce the number of local optima, and makes it less sensitive to different initialization.

## 4.2 Softmax Deep Deterministic Policy Gradients (SD2)

Here we present the design of SD2, which is short for Softmax Deep Deterministic Policy Gradients, where we build the softmax operator upon DDPG [31] with a single critic estimator.

Specifically, SD2 estimates the value function using the softmax operator, and the update of the critic of SD2 is defined by Eq. (2), where the the actor aims to optimize a soft estimation of the return.

$$\theta' = \theta + \alpha \mathbb{E}_{s,a \sim \rho} \left( r + \gamma \mathcal{T}_{\text{SD2}}(s') - Q(s,a;\theta) \right) \nabla_\theta Q(s,a;\theta). \tag{2}$$

In Eq. (2), $\mathcal{T}_{\text{SD2}}(s') = \text{softmax}_\beta(Q(s', \cdot; \theta^-))$. However, the softmax operator involves the integral, and is intractable in continuous action space. We express the Q-function induced by the softmax operator in expectation by importance sampling [18], and obtain an unbiased estimation by

$$\mathbb{E}_{a' \sim p} \left[ \frac{\exp(\beta Q(s', a'; \theta^-)) Q(s', a'; \theta^-)}{p(a')} \right] / \mathbb{E}_{a' \sim p} \left[ \frac{\exp(\beta Q(s', a'; \theta^-))}{p(a')} \right], \tag{3}$$

where $p(a')$ denotes the probability density function of a Gaussian distribution. In practice, we sample actions obtained by adding noises which are sampled from a Gaussian distribution $\epsilon \sim \mathcal{N}(0, \sigma)$ to the target action $\pi(s'; \phi^-)$, i.e., $a' = \pi(s'; \phi^-) + \epsilon$. Here, each sampled noise is clipped to $[-c, c]$ to ensure the sampled action is in $\mathcal{A}_c = [-c + \pi(s'; \phi^-), c + \pi(s'; \phi^-)]$. This is because directly estimating $\mathcal{T}_{\text{SD2}}(s')$ can incur large variance as $1/p(a')$ can be very large. Therefore, we limit the range of the action set to guarantee that actions are close to the original action, and that we obtain a robust estimate of the softmax Q-value. Due to space limitation, we put the full SD2 algorithm in Appendix C.1.

### 4.2.1 SD2 Reduces the Overestimation Bias

Besides the smoothing effect on the optimization landscape, we show in Theorem 3 that SD2 enables a better value estimation by reducing the overestimation bias in DDPG, for which it is known that the critic estimate can cause significant overestimation [15], where the proof is in Appendix C.2.

**Theorem 3** *Denote the bias of the value estimate and the true value induced by $\mathcal{T}$ as $\text{bias}(\mathcal{T}) = \mathbb{E}[\mathcal{T}(s')] - \mathbb{E}[Q(s', \pi(s'; \phi^-); \theta^{\text{true}})]$. Assume that the actor is a local maximizer with respect to the critic, then there exists noise clipping parameter $c > 0$ such that $\text{bias}(\mathcal{T}_{\text{SD2}}) \leq \text{bias}(\mathcal{T}_{\text{DDPG}})$.*

We validate the reduction effect in two MuJoCo [35] environments, Hopper-v2 and Walker2d-v2, where the experimental setting is the same as in Section 5. Figure 3 shows the performance comparison between DDPG and SD2, where the shaded area corresponds to standard deviation. The red horizontal line denotes the maximum return obtained by DDPG in evaluation during training, while the blue vertical lines show the number of steps for DDPG and SD2 to reach that score. As shown in Figure 3, SD2 significantly outperforms DDPG in sample efficiency and final

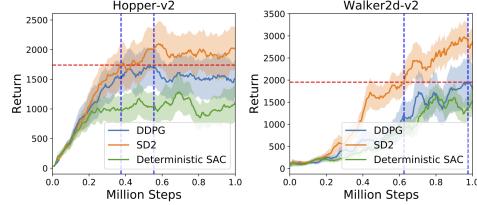

Figure 3: Performance comparison of DDPG and SD2, and Deterministic SAC.

performance. Estimation of value functions is shown in Figure 4(a), where value estimates are averaged over 1000 states sampled from the replay buffer at each timestep, and true values are estimated by averaging the discounted long-term rewards obtained by rolling out the current policy starting from the sampled states at each timestep. The bias of corresponding value estimates and true values is shown in Figure 4(b), where it can be observed that SD2 reduces overestimation and achieves a better estimation of value functions.

Regarding the softmax operator in SD2, one may be interested in comparing it with the log-sum-exp operator applied in SAC [19]. To study the effect of different operators, we compare SD2 with a variant of SAC with deterministic policy and single critic for fair comparison. The performance of Deterministic SAC (with fine-tuned parameter of log-sum-exp) is shown in Figure 3, which underperforms DDPG and SD2, where we also observe that its absolute bias is larger than that of DDPG, and worsens the overestimation problem. Its value estimates can be found in Appendix C.3.

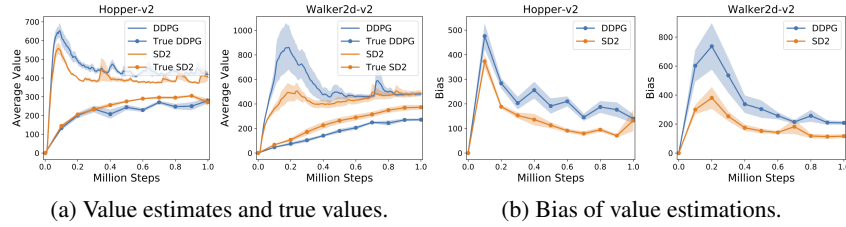

(a) Value estimates and true values.        (b) Bias of value estimations.

Figure 4: Comparison of estimation of value functions of DDPG and SD2.

## 4.3 Softmax Deep Double Deterministic Policy Gradients (SD3)

In Section 4.2.1, we have analyzed the effect of the softmax operator in the aspect of value estimation based on DDPG which suffers from overestimation. We now investigate whether the softmax operator is still beneficial when there is underestimation bias. We propose a novel method to leverage the softmax operator with double estimators, called S̲oftmax D̲eep D̲ouble D̲eterministic Policy Gradients (SD3). We show that SD3 enables a better value estimation in comparison with the state-of-the-art TD3 algorithm, which can suffer from a large underestimation bias.

TD3 [15] maintains a pair of critics as in Double Q-learning [20], which is a promising approach to alleviate overestimation in the value-based Q-learning [37] method. However, directly applying Double Q-learning still leads to overestimation in the actor-critic setting. To avoid the problem, Clipped Double Q-learning is proposed in TD3 [15], which clips the Q-value from the double estimator of the critic by the original Q-value itself. Specifically, TD3 estimates the value function by taking the minimum of value estimates from the two critics according to $y_1, y_2 = r + \gamma \min_{i=1,2} Q_i(s', \pi(s'; \phi^-); \theta_i^-)$. Nevertheless, it may incur large underestimation bias, and can affect performance [23, 10].

We propose to use the softmax operator based on double estimators to address the problem. It is worth noting that a direct way to combine the softmax operator with TD3, i.e., apply the softmax operator to the Q-value from the double critic estimator and then clip it by the original Q-value, as in Eq. (4) is ineffective.

$$y_i = r + \gamma \min \left( \mathcal{T}_{\mathrm{SD2}}^{-i}(s'), Q_i(s', \pi(s'; \phi^-); \theta_i^-) \right), \quad \mathcal{T}_{\mathrm{SD2}}^{-i}(s') = \mathrm{softmax}_\beta(Q_{-i}(s', \cdot; \theta_{-i}^-)). \quad (4)$$

This is because according to Theorem 3, we have $\mathcal{T}_{\mathrm{SD2}}^{-i}(s') \leq Q_{-i}(s', \pi(s'; \phi^-); \theta_{-i}^-)$, then the value estimates result in even larger underestimation bias compared with TD3. To tackle the problem, we propose to estimate the target value for critic $Q_i$ by $y_i = r + \gamma \mathcal{T}_{\mathrm{SD3}}(s')$, where

$$\mathcal{T}_{\mathrm{SD3}}(s') = \mathrm{softmax}_\beta \left( \hat{Q}_i(s', \cdot) \right), \quad \hat{Q}_i(s', a') = \min \left( Q_i(s', a'; \theta_i^-), Q_{-i}(s', a'; \theta_{-i}^-) \right). \quad (5)$$

Here, target actions for computing the softmax Q-value are obtained by the same way as in the SD2 algorithm in Section 4.2. The full SD3 algorithm is shown in Algorithm 1.

### 4.3.1 SD3 Improves the Underestimation Bias

In Theorem 4, we present the relationship between the value estimation of SD3 and that of TD3, where the proof is in Appendix D.1.

**Theorem 4** *Denote $\mathcal{T}_{\mathrm{TD3}}$, $\mathcal{T}_{\mathrm{SD3}}$ the value estimation functions of TD3 and SD3 respectively, then we have* $\mathrm{bias}(\mathcal{T}_{\mathrm{SD3}}) \geq \mathrm{bias}(\mathcal{T}_{\mathrm{TD3}})$.

As illustrated in Theorem 4, the value estimation of SD3 is larger than that of TD3. As TD3 leads to an underestimation value estimate [15], we get that SD3 helps to improve the underestimation bias of TD3. Therefore, according to our SD2 and SD3 algorithms, we conclude that the softmax operator can not only reduce the overestimation bias when built upon DDPG, but also improve the underestimation bias when built upon TD3. We empirically validate

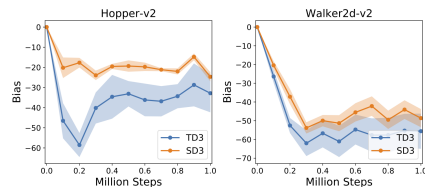

Figure 5: Comparison of the bias of value estimations of TD3 and SD3.

**Algorithm 1** SD3

1: Initialize critic networks $Q_1, Q_2$, and actor networks $\pi_1, \pi_2$ with random parameters $\theta_1, \theta_2, \phi_1, \phi_2$
2: Initialize target networks $\theta_1^- \leftarrow \theta_1, \theta_2^- \leftarrow \theta_2, \phi_1^- \leftarrow \phi_1, \phi_1^- \leftarrow \phi_1$
3: Initialzie replay buffer $\mathcal{B}$
4: **for** $t = 1$ to $T$ **do**
5:     Select action $a$ with exploration noise $\epsilon \sim \mathcal{N}(0, \sigma)$ based on $\pi_1$ and $\pi_2$
6:     Execute action $a$, observe reward $r$, new state $s'$ and done $d$
7:     Store transition tuple $(s, a, r, s', d)$ in $\mathcal{B}$ // $d$ is the done flag
8:     **for** $i = 1, 2$ **do**
9:         Sample a mini-batch of $N$ transitions $\{(s, a, r, s', d)\}$ from $\mathcal{B}$
10:         Sample $K$ noises $\epsilon \sim \mathcal{N}(0, \bar{\sigma})$
11:         $\hat{a}' \leftarrow \pi_i(s'; \phi_i^-) + \text{clip}(\epsilon, -c, c)$
12:         $\hat{Q}(s', \hat{a}') \leftarrow \min_{j=1,2} \left( Q_j(s', \hat{a}'; \theta_j^-) \right)$
13:         $\text{softmax}_\beta \left( \hat{Q}(s', \cdot) \right) \leftarrow \mathbb{E}_{\hat{a}' \sim p} \left[ \frac{\exp(\beta \hat{Q}(s', \hat{a}')) \hat{Q}(s', \hat{a}')}{p(\hat{a}')} \right] / \mathbb{E}_{\hat{a}' \sim p} \left[ \frac{\exp(\beta \hat{Q}(s', \hat{a}'))}{p(\hat{a}')} \right]$
14:         $y_i \leftarrow r + \gamma(1 - d)\text{softmax}_\beta \left( \hat{Q}(s', \cdot) \right)$
15:         Update the critic $\theta_i$ according to Bellman loss: $\frac{1}{N} \sum_s (Q_i(s, a; \theta_i) - y_i)^2$
16:         Update actor $\phi_i$ by policy gradient: $\frac{1}{N} \sum_s \left[ \nabla_{\phi_i}(\pi(s; \phi_i)) \nabla_a Q_i(s, a; \theta_i)|_{a=\pi(s; \phi_i)} \right]$
17:         Update target networks: $\theta_i^- \leftarrow \tau\theta_i + (1 - \tau)\theta_i^-, \phi_i^- \leftarrow \tau\phi_i + (1 - \tau)\phi_i^-$
18:     **end for**
19: **end for**

the theorem using the same two MuJoCo environments and estimation of value functions and true values as in Section 4.2.1. Comparison of the bias of value estimates and true values is shown in Figure 5, where the performance comparison is in Figure 8. As shown, SD3 enables better value estimations as it achieves smaller absolute bias than TD3, while TD3 suffers from a large underestimation bias. We also observe that the variance of value estimates of SD3 is smaller than that of TD3.

## 5   Experiments

In this section, we first conduct an ablation study on SD3, from which we aim to obtain a better understanding of the effect of each component, and to further analyze the main driver of the performance improvement of SD3. Then, we extensively evaluate the SD3 algorithm on continuous control benchmarks and compare with state-of-the-art methods.

We conduct experiments on continuous control tasks from OpenAI Gym [6] simulated by MuJoCo [35] and Box2d [8]. We compare SD3 with DDPG [24] and TD3 [15] using authors' open-sourced implementation [14]. We also compare SD3 against Soft Actor-Critic (SAC) [19], a state-of-the-art method that also uses double critics. Each algorithm is run with 5 seeds, where the performance is evaluated for 10 times every 5000 timesteps. SD3 uses double actors and double critics based on the structure of Double Q-learning [20], with the same network configuration as the default TD3 and DDPG baselines. For the softmax operator in SD3, the number of noises to sample $K$ is 50, and the parameter $\beta$ is mainly chosen from $\{10^{-3}, 5 \times 10^{-3}, 10^{-2}, 5 \times 10^{-2}, 10^{-1}, 5 \times 10^{-1}\}$ using grid search. All other hyperparameters of SD3 are set to be the same as the default setting for TD3 on all tasks except for Humanoid-v2, as TD3 with the default hyperparameters almost fails in Humanoid-v2. To better demonstrate the effectiveness of SD3, we therefore employ the fine-tuned hyperparameters provided by authors of TD3 [14] for Humanoid-v2 for DDPG, TD3 and SD3. Details for hyperparameters are in Appendix E.1, and the implementation details are publicly available at `https://github.com/ling-pan/SD3`.

### 5.1   Ablation Study

We first conduct an ablative study of SD3 in an MuJoCo environment HalfCheetah-v2 to study the effect of structure and important hyperparameters.

**Structure.** From Figure 6(a), we find that for SD3 and TD3, using double actors outperforms its counterpart with a single actor. This is because using a single actor as in TD3 leads to a same training target for both critics, which can be close during training and may not fully utilize the double

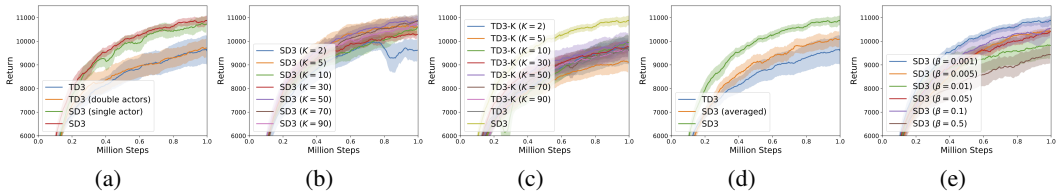

Figure 6: Ablation study on HalfCheetah-v2 (mean $\pm$ standard deviation). (a) Structure. (b) Number of noises $K$. (c) Comparison with TD3-$K$. (d) Comparison with SD3 (averaged). (e) Parameter $\beta$.

estimators. However, TD3 with double actors still largely underperforms SD3 (either with single or double actors).

**The number of noises $K$.** Figure 6(b) shows the performance of SD3 with varying number of noise samples $K$. The performance of all $K$ values is competitive except for $K = 2$, where it fails to behave stable and also underperforms other values of $K$ in sample efficiency. As SD3 is not sensitive to this parameter, we fix $K$ to be 50 in all environments as it performs best. Note that doing so does not incur much computation cost as setting $K$ to be 50 only takes 3.28% more runtime on average compared with $K = 1$ (in this case the latter can be viewed as a variant of TD3 with double actors).

**The effect of the softmax operator.** It is also worth studying the performance of a variant of TD3 using $K$ samples of actions to evaluate the Q-function (TD3-$K$). Specifically, TD3-$K$ samples $K$ actions by the same way as in SD3 to compute Q-values before taking the min operation (details are in Appendix E.2). As shown in Figure 6(c), TD3-$K$ outperforms TD3 for some large values of $K$, but only by a small margin and still underperforms SD3. We also compare SD3 with its variant SD3 (averaged) that directly averages the $K$ samples to compute the Q-function, which underperforms SD3 by a large margin as shown in Figure 6(d). Results confirm that the softmax operator is the key factor for the performance improvement for SD3 instead of other changes (multiple samples).

**The parameter $\beta$.** The parameter $\beta$ of the softmax operator directly affects the estimation of value functions, and controls the bias of value estimations, which is a critical parameter for the performance. A smaller $\beta$ leads to lower variance while a larger $\beta$ results in smaller bias. Indeed, there is an intermediate value that performs best that can best provide the trade-off as in Figure 6(e).

## 5.2 Performance Comparison

The performance comparison is shown in Figure 8, where we report the averaged performance as the solid line, and the shaded region denotes the standard deviation. As demonstrated, SD3 significantly outperforms TD3, where it achieves a higher final performance and is more stable due to the smoothing effect of the softmax operator on the optimization landscape and a better value estimation. Figure 7 shows the number of steps for TD3 and SD3 to reach the highest score of TD3 during training. We observe that SD3 learns much faster than TD3. It is worth noting that SD3 outperforms SAC in most environments except for Humanoid-v2, where both algorithms are competitive.

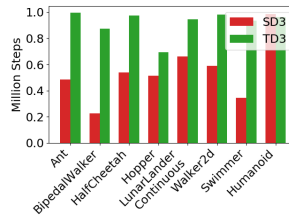

Figure 7: Sample efficiency comparison.

## 6 Related Work

How to obtain good value estimation is an important problem in reinforcement learning, and has been extensively investigated in discrete system control for deep Q-network (DQN) [20, 36, 3, 32, 23, 28]. Ensemble-DQN [3] leverages an ensemble of Q-networks which can reduce variance while Averaged-DQN [3] uses previously learned Q-value estimates by averaging them to lower value estimations. Lan et al. [23] propose to use an ensemble scheme to control the bias of value estimates for DQN [26]. In [32], Song et al. apply the softmax operator for discrete control in DQNs, and validate the performance gain by showing that softmax can reduce overestimation and gradient noise in DQN. In [28], Pan et al. propose a convergent variant of the softmax operator for discrete control. In this paper, our focus is to investigate the properties and benefits of the softmax operator in continuous control,

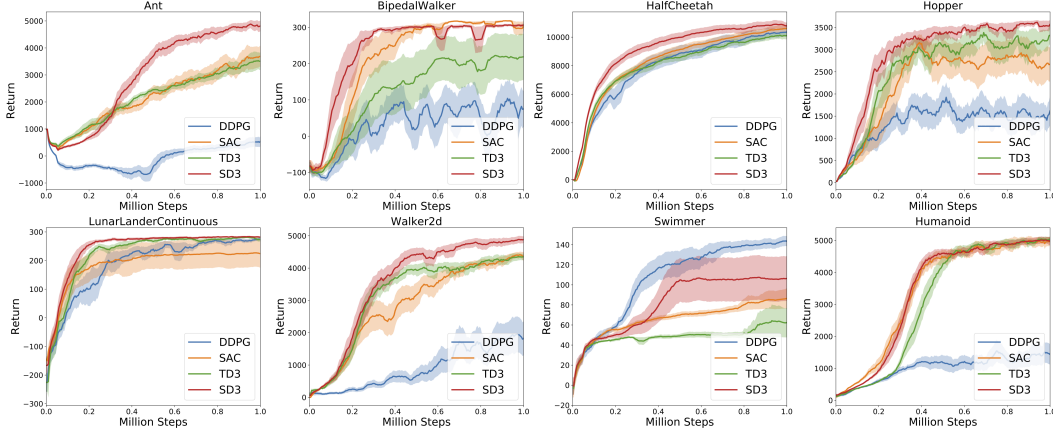

Figure 8: Performance comparison in MuJoCo environments.

where we provide new analysis and insights. TD3 [15] is proposed to tackle the overestimation problem in continuous action space. However, it can suffer from a large underestimation problem, which is a focus of this work. There are several works that build on and improve DDPG including prioritized experience replay [21], distributional [5], model-based [17, 7, 13], evolution methods [22], etc. Prior works [11, 27] generalize DDPG for learning a stochastic Gaussian policy, while we uncover and study benefits of softmax operator with deterministic policy. Achiam et al. [1] study the divergence problem of deep Q-learning, and propose PreQN to ensure that the value function update is non-expansive. However, PreQN can be computationally expensive, while SD3 is efficient.

## 7 Conclusion

In this paper, we show that it is promising to use the softmax operator in continuous control. We first provide a new analysis for the error bound between the value function induced by the softmax operator and the optimal in continuous control. We then show that the softmax operator (i) helps to smooth the optimization landscape, (ii) can reduce the overestimation bias and improve performance of DDPG when combined with single estimator (SD2), and (iii) can also improve the underestimation bias of TD3 when built upon double estimators (SD3). Extensive experimental results on standard continuous control benchmarks validate the effectiveness of the SD3 algorithm, which significantly outperforms state-of-the-art algorithms. For future work, it is interesting to study an adaptive scheduling of the parameter $\beta$ in SD2 and SD3. In addition, it also worths to quantify the bias reduction for overestimation and underestimation. It will also be an interesting direction to unify SD2 and SD3 into a same framework to study the effect on value estimations.

## Broader Impact

Recent years have witnessed unprecedented advances of deep reinforcement learning in real-world tasks involving high-dimensional state and action spaces that leverages the power of deep neural networks including robotics, transportation, recommender systems, etc. Our work investigates the Boltzmann softmax operator in updating value functions in reinforcement learning for continuous control, and provides new insights and further understanding of the operator. We show that the error bound of the value function under the softmax operator and the optimal can be bounded and it is promising to use the softmax operator in continuous control. We demonstrate the smoothing effect of the softmax operator on the optimization landscape, and shows that it can provide better value estimations. Experimental results show the potential of our proposed algorithm to improve final performance and sample efficiency. It will be interesting to apply our algorithm in practical applications.

## Acknowledgments and Disclosure of Funding

We thank the anonymous reviewers for their valuable feedbacks and suggestions. The work of Ling Pan and Longbo Huang was supported in part by the National Natural Science Foundation of China Grant 61672316, the Zhongguancun Haihua Institute for Frontier Information Technology and the Turing AI Institute of Nanjing.

## Footnotes

[1]See Appendix B.1 for the comparative study on the smoothing effect of TD3 and SD3.

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
