[Supplementary Material]

# Appendix for Softmax Deep Double Deterministic Policy Gradients

**Ling Pan[1], Qingpeng Cai[2], Longbo Huang[1]**
[1]Institute for Interdisciplinary Information Sciences, Tsinghua University
pl17@mails.tsinghua.edu.cn, longbohuang@tsinghua.edu.cn
[2]Alibaba Group
qingpeng.cqp@alibaba-inc.com

## A  Proofs in Section 4

### A.1  Proof of Theorem 1

**Theorem 1**  *Let $\mathcal{C}(Q, s, \epsilon) = \{a | a \in \mathcal{A}, Q(s,a) \geq \max_a Q(s,a) - \epsilon\}$ and $F(Q, s, \epsilon) = \int_{a \in \mathcal{C}(Q,s,\epsilon)} 1 da$ for any $\epsilon > 0$ and any state $s$. The difference between the max operator and the softmax operator satisfies*

$$0 \leq \max_a Q(s,a) - \text{softmax}_\beta(Q(s,\cdot)) \leq \frac{\int_{a \in \mathcal{A}} 1 da - 1 - \ln F(Q,s,\epsilon)}{\beta} + \epsilon. \quad (1)$$

*Proof.* For the left-hand-side, we have by definition that

$$\text{softmax}_\beta(Q(s,\cdot)) \leq \int_{a \in \mathcal{A}} \frac{\exp(\beta Q(s,a))}{\int_{a' \in \mathcal{A}} \exp(\beta Q(s,a')) da'} \max_{a'} Q(s,a') da \leq \max_a Q(s,a). \quad (2)$$

For the right-hand-side, we first provide a relationship between the softmax operator and the log-sum-exp operator $\text{lse}_\beta(Q(s,\cdot))$ (Eq. (9) in [4]) in continuous action spaces, i.e., $\text{lse}_\beta(Q(s,\cdot)) = \frac{1}{\beta} \ln \int_{a \in \mathcal{A}} \exp(\beta Q(s,a)) da$.

Denote the probability density function of the softmax distribution by $p_\beta(s,a) = \frac{\exp(\beta Q(s,a))}{\int_{a' \in \mathcal{A}} \exp(\beta Q(s,a')) da'}$. We have

$$\begin{aligned}
&\text{lse}_\beta(Q(s,\cdot)) - \text{softmax}_\beta(Q(s,\cdot)) \\
&= \frac{\ln \int_{a \in \mathcal{A}} \exp(\beta Q(s,a)) da}{\beta} - \int_{a \in \mathcal{A}} p_\beta(s,a) Q(s,a) da \\
&= \frac{\int_{a \in \mathcal{A}} p_\beta(s,a)(\ln \int_{a' \in \mathcal{A}} \exp(\beta Q(s,a')) da') da}{\beta} - \frac{\int_{a \in \mathcal{A}} p_\beta(s,a) \beta Q(s,a) da}{\beta} \\
&= \frac{\int_{a \in \mathcal{A}} -p_\beta(s,a) \ln p_\beta(s,a) da}{\beta}.
\end{aligned} \quad (3)$$

As $p_\beta(s,a)$ is non-negative, we have that $\forall a, -p_\beta(s,a) \ln p_\beta(s,a) \leq 1 - p_\beta(s,a)$.

Note that $\int_{a \in \mathcal{A}} p_\beta(s,a) da = 1$. We have

$$\text{lse}_\beta(Q(s,\cdot)) - \text{softmax}_\beta(Q(s,\cdot)) \leq \frac{\int_{a \in \mathcal{A}} 1 da - 1}{\beta}. \quad (4)$$

Secondly, by the definition of the log-sum-exp operator and the fact that $\mathcal{C}(Q, s, \epsilon)$ is a subset of $\mathcal{A}$, we have

$$
\begin{aligned}
\text{lse}_\beta\left(Q(s,\cdot)\right) &= \frac{\ln \int_{a\in\mathcal{A}} \exp(\beta Q(s,a))da}{\beta} \\
&\geq \frac{\ln \int_{a\in\mathcal{C}(Q,s,\epsilon)} \exp(\beta Q(s,a))da}{\beta} \\
&\geq \frac{\ln \int_{a\in\mathcal{C}(Q,s,\epsilon)} \exp\left(\beta(\max_a Q(s,a) - \epsilon)\right)da}{\beta} \\
&\geq \frac{\ln F(Q,s,\epsilon) + \beta\left(\max_a Q(s,a) - \epsilon\right)}{\beta}.
\end{aligned}
\tag{5}
$$

As a result, we get the inequality of the max operator and the log-sum-exp operator as in Eq. (6).

$$
\text{lse}_\beta\left(Q(s,\cdot)\right) \geq \max_a Q(s,a) + \frac{\ln F(Q,s,\epsilon) - \beta\epsilon}{\beta}.
\tag{6}
$$

Finally, combining Eq. (4) and Eq. (6), we obtain Eq. (1). $\qquad\square$

**Remark.** In a special case where for any state $s$, the set of the maximum actions at $s$ with respect to the Q-function $Q(s,a)$ covers a continuous set, $\epsilon$ can be $0$ and the upper bound in Eq. (1) still holds as $F(Q,s,0) > 0$, which turns into $\frac{\int_{a\in\mathcal{A}} 1da - 1 - \ln F(Q,s,0)}{\beta}$. Please also note that when $Q(s,a)$ is a constant function w.r.t. $a$, the upper bound will be $0$ in this case as from Eq. (3), we get that $\text{lse}_\beta\left(Q(s,\cdot)\right) - \text{softmax}_\beta\left(Q(s,\cdot)\right) = \frac{\ln \int_{a\in\mathcal{A}} 1da}{\beta}$ and $\text{lse}_\beta\left(Q(s,\cdot)\right) \geq \max_a Q(s,a) + \frac{\ln \int_{a\in\mathcal{A}} 1da}{\beta}$.

### A.1.1 Discussion of Theorem 1 and Results in [6].

A recent work [6] studies the distance between the max and the softmax operators in the discrete setting. However, the proof technique in [6] is limited to discrete action space, and cannot be naturally extended to continuous action space. Specifically, following the line of analysis in [6], the gap between the max operator and the softmax operator is given by

$$
\frac{\int_{a\in\mathcal{A}} \frac{1}{D} \exp(-\beta\delta(s,a))\delta(s,a)da}{\int_{a\in\mathcal{A}} \frac{1}{D} \exp(-\beta\delta(s,a))da} = \frac{\int_{a\in\mathcal{A}-\mathcal{A}_m} \frac{1}{D} \exp(-\beta\delta(s,a))\delta(s,a)da}{c + \int_{a\in\mathcal{A}-\mathcal{A}_m} \frac{1}{D} \exp(-\beta\delta(s,a))da},
\tag{7}
$$

where $\mathcal{A}_m$ is the set of actions where the Q-function $Q(s,\cdot)$ attains the maximum value, $\delta(s,a) = \max_{a'} Q(s,a') - Q(s,a)$, $D = \int_{a\in\mathcal{A}} 1da$, and $c = \int_{a\in\mathcal{A}} \frac{I_{a\in\mathcal{A}_m}}{D}da$, where $I_{a\in\mathcal{A}_m}$ is the indicator function of event $\{a\in\mathcal{A}_m\}$. Please note that the analysis in [6] requires that $c > 0$, which does not always hold in the continuous case. As a result, the proof technique in [6] cannot be naturally extended to the continuous action setting, and we provide a new and different analysis in Theorem 1.

### A.2 Proof of Theorem 2

**Theorem 2** *For any iteration t, the difference between the optimal value function $V^*$ and the value function induced by softmax value iteration at the t-th iteration $V_t$ satisfies:*

$$
||V_t - V^*||_\infty \leq \gamma^t ||V_0(s) - V^*(s)||_\infty + \frac{1}{1-\gamma}\frac{\beta\epsilon + \int_{a\in\mathcal{A}} 1da - 1}{\beta} - \sum_{k=1}^{t} \gamma^{t-k}\frac{\min_s \ln F(Q_k, s, \epsilon)}{\beta}.
\tag{8}
$$

*Proof.* By the definition of softmax value iteration, we get

$$
\begin{aligned}
&|V_{t+1}(s) - V^*(s)| \\
=&|\text{softmax}_\beta\left(Q_{t+1}(s,\cdot)\right) - \max_a Q^*(s,a)| \\
\leq&|\text{softmax}_\beta\left(Q_{t+1}(s,\cdot)\right) - \max_a Q_{t+1}(s,a)| + |\max_a Q_{t+1}(s,a) - \max_a Q^*(s,a)|.
\end{aligned}
\tag{9}
$$

According to Theorem 1 and the fact that the max operator is non-expansive [5], we have

$$|V_{t+1}(s) - V^*(s)| \leq \frac{\beta\epsilon + \int_{a\in\mathcal{A}} 1 da - 1 - \ln F(Q_{t+1}, s, \epsilon)}{\beta} + \max_a |Q_{t+1}(s,a) - Q^*(s,a)|. \tag{10}$$

We also have the following inequality

$$|Q_{t+1}(s,a') - Q^*(s,a')| \leq \gamma \max_{s'} |V_t(s') - V^*(s')|. \tag{11}$$

Combining (10) and (11), we obtain

$$||V_{t+1}(s) - V^*(s)||_\infty \leq \frac{\beta\epsilon + \int_{a\in\mathcal{A}} 1 da - 1 - \min_s \ln F(Q_{t+1}, s, \epsilon)}{\beta} + \gamma ||V_t(s) - V^*(s)||_\infty. \tag{12}$$

Therefore, we have

$$
\begin{aligned}
&||V_t(s) - V^*(s)||_\infty \\
&\leq \gamma^t ||V_0(s) - V^*(s)||_\infty + \sum_{k=1}^{t} \gamma^{t-k} \frac{\beta\epsilon + \int_{a\in\mathcal{A}} 1 da - 1 - \min_s \ln F(Q_k, s, \epsilon)}{\beta} \\
&\leq \gamma^t ||V_0(s) - V^*(s)||_\infty + \frac{1}{1-\gamma} \frac{\beta\epsilon + \int_{a\in\mathcal{A}} 1 da - 1}{\beta} - \sum_{k=1}^{t} \gamma^{t-k} \frac{\min_s \ln F(Q_k, s, \epsilon)}{\beta}.
\end{aligned}
\tag{13}
$$

$\square$

## B  Softmax Helps to Smoooth the Optimization Landscape

### B.1  Comparative Study on the Smoothing Effect of TD3 and SD3

We demonstrate the smoothing effect of SD3 on the optimization landscape in this section, where experimental setup is the same as in Section 4.1 in the text for the comparative study of SD2 and DDPG. Experimental details can be found in Section B.2.

(a) Performance comparison.        (b) Scatter plot.

Figure 1: Analysis of smoothing effect of TD3 and SD3 in the MoveCar environment.

The performance comparison of SD3 and TD3 is shown in Figure 1(a), where SD3 significantly outperforms TD3. Next, we analyze the smoothing effect of SD3 on the optimization landscape by the same way as in Section 4.1 in the text. We obtain the scatter plot of random perturbation in Figure 1(b). Specifically, $\phi_0$ and $\phi_1$ are policy parameters learned by TD3 and SD3 during training from a same random initialization, where $\phi_0$ corresponds to a sub-optimal policy that determines to move to the right at the initial position $x_0$ while $\phi_1$ corresponds to an optimal policy that determines to move to the left and is able to stay in the high-reward region. As demonstrated in Figure 1(b), for TD3, we observe that blue triangles are around the origin, so the perturbations for its policy parameters $\phi_0$ are close to zero. This implies that there is a flat region in $\mathcal{L}_{TD3}$ and can be difficult

for gradient-based methods to escape from [1]. For SD3, as the perturbations for its policy parameters $\phi_1$ with respect to $\mathcal{L}_{\text{SD3}}$ are all positive (orange circles), the point $\phi_1$ is likely a local optimum. To demonstrate the critical effect of the softmax operator on the objective, we also evaluate the change of the loss function $\mathcal{L}_{\text{SD3}}$ of SD3 with respect to parameters $\phi_0$ from TD3. As shown in Figure 1(b), the green squares indicate that the loss $\mathcal{L}_{\text{SD3}}$ can be reduced following many directions, which shows the advantage of optimizing the softmax version of the objective.

(a) Performance comparison.      (b) Scatter plot.

Figure 2: Analysis of smoothing effect of TD3-K and SD3 in the MoveCar environment.

So far, we have demonstrated the smoothing effect of SD3 over TD3. We further compare SD3 and TD3-$K$ (which is introduced in Section 5.1 in the text) to demonstrate the critical effect of the softmax operator on the objective. The performance comparison is shown in Figure 2(a), where $K$ is the same as in SD3. As shown, although TD3-$K$ performs better than TD3, SD3 still outperforms TD3-$K$ by a large margin in final performance. With the same way of random perturbation as in the previous part, we demonstrate the scatter plot of the random perturbation in Figure 2(b). Similarly, the evaluation of the change of the loss function $\mathcal{L}_{\text{SD3}}$ of SD3 with respect to parameters $\phi_0$ from TD3-$K$ (green squares) shows the critical effect of the softmax operator on the objective, as the loss can be reduced following a number of directions.

## B.2 Experimental Setup in the MoverCar Environment

Hyperparameters of DDPG and SD2 are summarized in Table 1. For TD3 and SD3, hyperparameters are the same as in Table 2, except that we use $\mathcal{N}(0, 0.5)$ as in Table 1 for exploration to ensure that the exploration noise is high enough during training to exclude the effect of lack of exploration. For the TD3-$K$ algorithm in Figure 2, $K$ is the same as in SD3. We run all algorithms are 100 times (with different random seeds 0-99).

Table 1: Hyperparameters of DDPG and SD2.

| Hyperparameter | Value |
|---|---|
| Shared hyperparameters (From [3]) | |
|     Batch size | 100 |
|     Critic network | $(400, 300)$ |
|     Actor network | $(400, 300)$ |
|     Learning rate | $10^{-3}$ |
|     Optimizer | Adam |
|     Replay buffer size | $10^6$ |
|     Warmup steps | $10^4$ |
|     Exploration policy | $\mathcal{N}(0, 0.5)$ |
|     Discount factor | 0.99 |
|     Target update rate | $5 \times 10^{-3}$ |
| Hyperparameters for SD2 | |
|     Number of samples $K$ | 50 |
|     Action sampling noise $\bar{\sigma}$ | 0.2 |
|     Noise clipping coefficient $c$ | 0.5 |

# C Softmax Deep Deterministic Policy Gradients

## C.1 The SD2 Algorithm

The full SD2 algorithm is shown in Algorithm 1.

---

**Algorithm 1** SD2

---

1: Initialize the critic network $Q$ and the actor network $\pi$ with random parameters $\theta, \phi$
2: Initialize target networks $\theta^- \leftarrow \theta$, $\phi^- \leftarrow \phi$
3: Initialzie replay buffer $\mathcal{B}$
4: **for** $t = 1$ to $T$ **do**
5:      Select action $a$ with exploration noise $\epsilon \sim \mathcal{N}(0, \sigma)$ based on $\pi$
6:      Execute action $a$, observe reward $r$, new state $s'$ and done $d$
7:      Store transition tuple $(s, a, r, s', d)$ in $\mathcal{B}$
8:      Sample a mini-batch of $N$ transitions $\{(s, a, r, s', d)\}$ from $\mathcal{B}$
9:      Sample $K$ noises $\epsilon \sim \mathcal{N}(0, \bar{\sigma})$
10:      $\hat{a}' \leftarrow \pi(s'; \phi^-) + \text{clip}(\epsilon, -c, c)$
11:      $\text{softmax}_\beta \left( Q(s', \cdot; \theta^-) \right) \leftarrow \mathbb{E}_{\hat{a}' \sim p} \left[ \frac{\exp(\beta Q(s', \hat{a}'; \theta^-)) Q(s', \hat{a}'; \theta^-)}{p(\hat{a}')} \right] / \mathbb{E}_{\hat{a}' \sim p} \left[ \frac{\exp(\beta Q(s', \hat{a}'; \theta^-))}{p(\hat{a}')} \right]$
12:      $y_i \leftarrow r + \gamma(1 - d)\text{softmax}_\beta \left( Q(s', \cdot; \theta^-) \right)$
13:      Update the parameter $\theta$ of the critic according to Bellman loss: $\frac{1}{N} \sum_s (Q(s, a; \theta) - y)^2$
14:      Update the parameter $\phi$ of the actor by policy gradient: $\frac{1}{N} \sum_s \left[ \nabla_\phi (\pi(s; \phi)) \nabla_a Q(s, a; \theta)|_{a = \pi(s; \phi)} \right]$
15:      Update target networks: $\theta^- \leftarrow \tau\theta + (1 - \tau)\theta^-$, $\phi^- \leftarrow \tau\phi + (1 - \tau)\phi^-$
16: **end for**

---

## C.2 Proof of Theorem 3

**Theorem 3** *Denote the bias of the value estimate and the true value induced by $\mathcal{T}$ as $\text{bias}(\mathcal{T}) = \mathbb{E}[\mathcal{T}(s')] - \mathbb{E}[Q(s', \pi(s'; \phi^-); \theta^{\text{true}})]$. Assume that the actor is a local maximizer with respect to the critic, then there exists noise clipping parameter $c > 0$ such that $\text{bias}(\mathcal{T}_{\text{SD2}}) \leq \text{bias}(\mathcal{T}_{\text{DDPG}})$.*

*Proof.* By definition, we have

$$\mathcal{T}_{\text{DDPG}}(s') = Q(s', \pi(s'; \phi^-); \theta^-), \quad \mathcal{T}_{\text{SD2}}(s') = \text{softmax}_\beta(Q(s', \cdot; \theta^-)) \tag{14}$$

Assume that the actor is a local maximizer with respect to the critic. There exists $c > 0$ such that for any state $s'$, the action selected by the policy $\pi(s'; \phi^-)$ at state $s'$ is a local maximum of the Q-function $Q(s', \cdot; \theta^-)$, i.e.,

$$Q(s', \pi(s'; \phi^-); \theta^-) = \max_{a \in \mathcal{A}_c} Q(s', a; \theta^-). \tag{15}$$

From Theorem 1, we have that

$$\text{softmax}_\beta(Q(s', \cdot; \theta^-)) \leq \max_{a \in \mathcal{A}_c} Q(s', a; \theta^-). \tag{16}$$

Thus,

$$\text{softmax}_\beta(Q(s', \cdot; \theta^-)) \leq Q(s', \pi(s'; \phi^-); \theta^-), \tag{17}$$

and we obtain $\mathcal{T}_{\text{SD2}}(s') \leq \mathcal{T}_{\text{DDPG}}(s')$. Therefore, $\text{bias}(\mathcal{T}_{\text{SD2}}) \leq \text{bias}(\mathcal{T}_{\text{DDPG}})$. □

## C.3 Comparison of Estimation of Value Functions of DDPG and Deterministic SAC

Figure 3(a) shows the value estimates and true values of DDPG and Deterministic SAC, and Figure 3(b) demonstrates the corresponding bias of value estimations, from which we observe that Deterministic SAC incurs larger overestimation bias in comparison with DDPG.

(a) Value estimates and true values.  (b) Bias of value estimations.

Figure 3: Comparison of estimations of value functions of DDPG and Deterministic SAC.

# D  Softmax Deep Double Deterministic Policy Gradients

## D.1  Proof of Theorem 4

**Theorem 4** *Denote* $\mathcal{T}_{\text{TD3}}$, $\mathcal{T}_{\text{SD3}}$ *the value estimation functions of TD3 and SD3 respectively, then we have* $\text{bias}(\mathcal{T}_{\text{SD3}}) \geq \text{bias}(\mathcal{T}_{\text{TD3}})$.

*Proof.* By definition, we have

$$\mathcal{T}_{\text{TD3}}(s') = \hat{Q}_i(s', \hat{a}'), \quad \mathcal{T}_{\text{SD3}}(s') = \text{softmax}_\beta(\hat{Q}_i(s', \hat{a}')). \tag{18}$$

Since

$$\mathbb{E}\left[\hat{Q}_i(s', \hat{a}')\right] = \mathbb{E}\left[\text{softmax}_0(\hat{Q}_i(s', \hat{a}'))\right], \tag{19}$$

it suffices to prove that $\forall \beta \geq 0$,

$$\text{softmax}_\beta\left(\hat{Q}_i(s', \hat{a}')\right) \geq \text{softmax}_0\left(\hat{Q}_i(s', \hat{a}')\right). \tag{20}$$

Now we show that the softmax operator is increasing with $\beta$. By definition,

$$
\begin{aligned}
&\nabla_\beta \text{softmax}_\beta \left(Q(s, \cdot)\right) \\
=&\nabla_\beta \frac{\int_{a \in \mathcal{A}} e^{\beta Q(s,a)} Q(s,a) da}{\int_{a' \in \mathcal{A}} e^{\beta Q(s,a')} da'} \\
=&\frac{\int_{a \in \mathcal{A}} e^{\beta Q(s,a)} Q^2(s,a) da \times \int_{a' \in \mathcal{A}} e^{\beta Q(s,a')} da'}{\left(\int_{a' \in \mathcal{A}} e^{\beta Q(s,a')} da'\right)^2} - \frac{\left(\int_{a \in \mathcal{A}} e^{\beta Q(s,a)} Q(s,a) da\right)^2}{\left(\int_{a' \in \mathcal{A}} e^{\beta Q(s,a')} da'\right)^2}.
\end{aligned}
\tag{21}
$$

From the Cauchy-Schwarz inequality, we have $\forall \beta, \nabla_\beta \text{softmax}_\beta\left(Q(s, \cdot)\right) \geq 0$. Thus, $\text{softmax}_\beta$ attains its minimum at $\beta = 0$. $\square$

# E  Experimental Setup

## E.1  Hyperparameters

Hyperparameters of DDPG, TD3, and SD3 are shown in Table 2, where DDPG refers to 'OurDDPG' in [2]. Note that all hyperparameters are the same for all environments except for Humanoid-v2, as TD3 with default hyperparameters in this environment almost fails. For Humanoid-v2, the hyperparameters is a tuned set as provided in author's open-source implementation [2] to make TD3 work in this environment. For SD3, the parameter $\beta$ is $10^{-3}$ for Ant-v2, $5 \times 10^{-3}$ for HalfCheetah-v2, $5 \times 10^{-2}$ for BipedalWalker-v2, Hopper-v2, and Humanoid-v2, $10^{-1}$ for Walker2d-v2, $5 \times 10^{-1}$ for LunarLanderContinuous-v2, and a relatively large $\beta = 5 \times 10^2$ for Swimmer-v2.

Table 2: Hyperparameters of DDPG, TD3, and SD3.

| Hyperparameter | All environments except for Humanoid-v2 | Humanoid-v2 |
|---|:---:|:---:|
| Shared hyperparameters (From [3, 2]) | | |
|     Batch size | 100 | 256 |
|     Critic network | $(400, 300)$ | $(256, 256)$ |
|     Actor network | $(400, 300)$ | $(256, 256)$ |
|     Learning rate | $10^{-3}$ | $3 \times 10^{-4}$ |
|     Optimizer | Adam | |
|     Replay buffer size | $10^6$ | |
|     Warmup steps | $10^4$ | |
|     Exploration policy | $\mathcal{N}(0, 0.1)$ | |
|     Discount factor | 0.99 | |
|     Target update rate | $5 \times 10^{-3}$ | |
|     Noise clip | 0.5 | |
| Hyperparameters for TD3 (From [3]) | | |
|     Target update interval | 2 | |
|     Target noise | 0.2 | |
| Hyperparameters for SD3 | | |
|     Number of samples $K$ | 50 | |
|     Action sampling noise $\bar{\sigma}$ | 0.2 | |

## E.2   The TD3-$K$ Algorithm

We compare SD3 with TD3-$K$, a variant of TD3 that uses $K$ samples of actions to evaluate the Q-function, to demonstrate that using multiple samples is not the main factor for the performance improvement of SD3. Specifically, TD3-$K$ samples $K$ actions $a'$ by the same way as in SD3 to compute Q-values, and take the $\min$ operation over the averaged Q-values, i.e., $y_{1,2} = r + \gamma \min_{i=1,2} \left( \frac{1}{K} \sum_{j=1}^{K} Q_i(s', a'; \theta_i^-) \right)$.