[Reviews · NeurIPS 2020]

Review 1

Summary and Contributions: This paper proposes to use the softmax operator, instead of max, in DDPG/TD3 to correct the overestimation/underestimation bias. Practical algorithms SD2/SD3 are proposed with good empirical studies. In particular, it is empirically shown that the softmax operator helps smooth the loss landscape, and the proposed methods achieve state-of-the-art results on MuJoCo and Box2d. ===========After rebuttal========= I have read the other reviews and the rebuttal. I don't think the rebuttal address my questions in the original review and I still have some concern in the rigorous analysis and the mathematical writing of the paper. On the other hand, the idea of the paper and its empirical analysis are interesting. I will keep my original score.

Strengths: (1) It is shown on a simple example that the softmax operator helps smooth the loss landscape, thus help in the optimization. It is also empirically shown that the softmax operator reduces the overestimation/underestimation problem compared to DDPG/TD3. An ablation study is also included in the paper. Overall, the empirical investigation of the proposed algorithms in this paper is careful and detailed. (2) This paper proposes a practical approximation method for estimating the softmax operator, as it includes integration in the definition of the softmax operator. (3) The proposed algorithms achieve state-of-the-art results.

Weaknesses: 1) There is no analysis of how the softmax can help smooth the loss landscape. (2) Theorem 3 and Theorem 4 seem to be crude. In Theorem 3, while SD2 helps to reduce the overestimation bias, the bias of T_{SD2} could still be larger as there is no guarantee that SD2 would not underestimate the value. Theorem 4 seems to be flawed. In particular, in the proof of theorem 4, why would Equation (19) hold? Note that softmax_0 is generally taking the average Q value.

Correctness: (1) In example 1, the rate is correct only if \beta is large. If \beta is small, the rate is constant. (2) Right after theorem 2, what if lnF is negative? (2) The proofs of theorem 3 and theorem 4 may need some corrections.

Clarity: Overall the paper is easy to follow. The structure is clear and the experiments are clearly described. Some notations are used in the paper without definitions. (1) Any condition on the action set A in the paper? (2) In theorem 3/4, what is the definition of bias? (3) Line 231, "y_i = r + \gamma min( ****)" seems to have a typo (4) What is the definition of \Tau_{SD3}?

Relation to Prior Work: Yes.

Reproducibility: Yes

Additional Feedback: In section 4.2, a pratical approximation method is proposed in estimating the softmax operator. How is the performance of this method?


Review 2

Summary and Contributions: The work provide strong theoretical, analytical, and empirical argument for an improved way to bootstrap values from multiple critics using soft-max as opposed to a cold max operator. A key technique is the use of multiple actors.

Strengths: This paper is simply a pleasure to read. The motivate is quite clear and well-founded, supported by cited references. The method section involve a simple toy domain that illustrates the benefit quite well. The experimental result section shows clear results and thorough discussion over a complete set of standard test domains, providing much confidence that this method would be impactful and should be adopted as a replacement of TD3 for continuous control. The change is simple, well-motivated, and clearly supported by the evidence.

Weaknesses: I wonder if there is a more fundamental view why soft-max alieviates both the over estimation problem *and* the under estimation problem. If the logits are participating in this weighted averaging in a subtle, but important fashion, such as being an energy model. It would also be good to relate this more broadly in the related works session to ensemble methods. For me personally that connection would be interesting.

Correctness: Yes.

Clarity: It is a pleasure to read.

Relation to Prior Work: Yes, very well motivated.

Reproducibility: Yes

Additional Feedback: I came out quite inspired by this work. Thank you for writing this paper! - It would help a bit if the text features that SD3 uses double actors a bit more prominently. During my reading, it was note really clear that SD3 uses two actors up untill the experiment section, when the comparison took place. - Although it is clear to practitioners, sometimes a good highlevel figure for the algorithm would help with illustrating the high-level structure, and differences against related methods. I understand that this is might be difficulty given the limited space for submission. - Some of the fonts in the figures are hard to read.


Review 3

Summary and Contributions: This paper analyses the use of a softmax operator applied to the Q-target of a continuous control RL algorithms such as DPG. It provides some theory (bounding the approximation error), insight (smoothing of optimisation landscape and reduction of overoptimism), and performance (outperforming state-of-the-art TD3 on standard control suite tasks).

Strengths: This is a nice paper. The paper is packed with small insightful experiments. It's great to see a careful study that unpicks the contributions of a simple idea to understand its benefits, and then puts it back together in a logical way to arrive at improved overall performance in standard benchmark tasks.

Weaknesses: The paper is what it is - a clear but narrow contribution based upon the introduction of softmax operators into existing algorithms.

Correctness: There were a couple of points I was unclear on: -The bounds include an integral over the action space. Presumably this then requires that the action space is bounded? I didn't see this stipulated. Nor is it clear to me that for large action spaces that the approximation bounds are necessarily very meaningful. What am I missing? -I was unclear about some terminology. Does the c > 0 in Theorem 3 refer to the noise clipping in the previous paragraph? Is it then specific to this particular form? -What is the (1-d) term in the main algorithm box? -Section 4.2 mentions importance sampling. But the algorithm appears to simply apply the softmax to sampled actions, without any importance sampling that I can see. I think I'm missing something here - what is actually going on?

Clarity: The paper is generally well written. Section 4.2 is too dense to fully understand, and this led to most of my questions and possible misunderstandings. It's a shame not to have the main algorithm box in main text.

Relation to Prior Work: Yes this is clear

Reproducibility: Yes

Additional Feedback:


Review 4

Summary and Contributions: This draft extended the idea of using the softmax Bellman operator in the discrete action space to the continuous case. The authors showed both the performance bound as well as the overestimation bias reduction. Empirical results were presented to demonstrate the superior performance of the proposed algorithms.

Strengths: 1. Extending the idea of averaging the Q-functions to the policy gradient methods is novel. 2. The author provided enough empirical evidence to support the claim, including the plots of reward and bias, which is indeed helpful to understand the paper. 3. The performance on the MuJoCo environment is encouraging as it consistently outperforms the previous method.

Weaknesses: I am a bit concerned about the significance of some theoretical results: In Theorem 1, the bound depends on \epsilon, even though \beta goes to infinity. Similarly in Theorem 2, the error converges to a non-zero value. These results are a bit surprising, given that the theorem in Song et al. showed it converges to zero. What would be the factors to lead to this gap?

Correctness: Yes

Clarity: This paper is well written.

Relation to Prior Work: This paper discussed clearly how it is different from previous work.

Reproducibility: Yes

Additional Feedback: I have the following questions for the authors to clarify and respond. 1. For the bias definition in Theorems 3 and 4, is E [T (s')] also dependent on \theta^{true}? If yes, would this be a reasonable assumption? 2. The authors showed that the proposed estimator can simultaneously reduce over- and under-estimation bias. Such results, however, definitely depend on the choice of \beta. Could you elaborate more on how to choose this parameter? ===After Rebuttal === Thanks for the clarification. I am a bit concerned that the bounds in Theorems 1 and 2 may not be tight enough, which has also been shared by other reviewers. For instance, if Q(s, a) is set to be a constant w.r.t. a in Theorem 1, max == softmax and C(Q, s, \epsilon) = A. However, the upper bound there goes to a non-trivial positive value. That being said, I still see the practical benefits of this submission, and hope the authors can address the issue on theoretical rigor.

[Author Response · NeurIPS 2020]

**To Reviewer #3**: Thank you for your careful reading and thoughtful reviews.

*Q1: Theorems 3 and 4.* (i) Theorem 3: Theorem 3 shows that SD2 helps to reduce the overestimation bias compared
with DDPG. We empirically show that SD2 does not underestimate and can reduce the absolute bias in Figure 4. It will
be interesting to further study the theoretical problem in future work. (ii) Clarification of Theorem 4: We clarify the
correctness of Theorem 4 as below. The left-hand side in Eq. (19) equals to $\mathbb{E}\left[\mathcal{T}_{\text{TD3}}(s')\right]$. Since TD3 uses target policy
smoothing (which adds a sampled noise to the action) when estimating the value of $s'$ in implementation, $\mathbb{E}\left[\mathcal{T}_{\text{TD3}}(s')\right]$
is exactly the averaged Q-value and Eq. (19) holds.

*Q2: The rate in example 1.* We present example 1 to show that our bound is tight when $\beta$ is large. We will clarify this in
the revised version to make this point clear.

*Q3: What if $\ln F$ is negative in Theorem 2?* In the case where $\ln F$ is negative, the last term on the right-hand side of
Theorem 2 still converges to $0$, as $\ln F$ is bounded for any given positive $\epsilon$. Thus, the error between the value function
induced by the softmax operator and the optimal one can still be bounded and controlled.

*Q4: Clarifications of condition and definitions.* (i) Condition on the action set $\mathcal{A}$. It is required to be bounded, and we
will clarify this point in the paper. (ii) Definition of bias. As defined in line 188 in the text, it denotes the difference
between the estimated value of the next state induced by the operator $\mathcal{T}$ and the true value of the next state. (iii)
Definition of $\mathcal{T}_{\text{SD3}}$. The definition is given in Eq. (18) in the appendix, and we will formally define it in the main text.

*Q5: How is the performance of the proposed approximation method?* The results in continuous
control tasks can validate that the proposed practical approximation method achieves good
performance. We also conduct an ablative experiment which studies the effect of noise clipping
that we proposed for a robust estimate of the softmax Q-value in HalfCheetah-v2. As shown
in the figure, SD3 outperforms its counterpart without using noise clipping as expected. We
will discuss it in the paper.

**To Reviewer #4**: Thank you for your careful reading and valuable comments, and we greatly appreciate your sugges-
tions! We will clarify the details, include a high-level figure for the structure, and polish the figures to make the fonts
larger in the revised version.

*Q1: A more fundamental view why softmax alleviates both over- and under-estimation problem.* We appreciate the
suggestion, and it is an interesting direction to unify SD2 and SD3 into a same framework that leverages softmax and
single or double critics to study the effects on value estimations. We will try to further investigate it in future research.

*Q2: Related works about ensemble methods.* Thanks for the suggestion, we will definitely incorporate the discussion
and connection with ensemble methods in the paper.

**To Reviewer #6**: Thank you for your detailed evaluation of our paper and thoughtful reviews, and the comments are
greatly appreciated! We will restructure Section 4.2 to make it more clear.

*Q1: About the action space.* (i) Requirement: Yes, the action space is required to be bounded. Thanks for pointing this
out, and we will clarify this in the paper. (ii) Large action space: For large action space, the gap will also approach to
$\epsilon/(1-\gamma)$ as $\beta$ increases. It will be an interesting direction to further improve the theoretical bound.

*Q2: Clarifications of notations and the algorithm.* Thanks for pointing these out, and we will clarify them in the revised
version. (i) The term $c$ in Theorem 3. Yes, it refers to the noise clipping in Section 4.2 (line 179), based on which we
defined the SD2 operator. Theorem 3 proves that the SD2 operator defined in the paper with this form helps to reduce
the overestimation bias compared with DDPG. (ii) The $(1-d)$ term in the algorithm box. The notation $d$ refers to the
boolean type done signal, i.e., whether the step is the end of an episode. (iii) Importance sampling in the algorithm box.
We will elaborate the details for computing softmax with importance sampling in the algorithm box.

**To Reviewer #7**: Thank you for your careful reading and thoughtful reviews.

*Q1: Clarification of the significance of Theorems 1 and 2.* Thanks for the question. The reason why the theorem in Song
et al. shows that the bound converges to $0$ (which considers the discrete case) while the bound in our paper does not, is
due to the critical difference between continuous and discrete action spaces, where we have discussed the difference in
Appendix A.1.1. We show that for any $\epsilon > 0$, our bound can converge to $\epsilon/(1-\gamma)$, which can be arbitrarily close to $0$.

*Q2: Does the first term in the bias definition depends on $\theta^{\text{true}}$?* Please note that $\mathbb{E}\left[\mathcal{T}(s')\right]$ is determined by the target
policy network and the target value network with parameter $\theta^-$, and does not depend on $\theta^{\text{true}}$.

*Q3: How to choose the parameter $\beta$?* For implementation, we use grid search to find the best value of $\beta$ to trade-off
between the bias and variance of value estimates, as discussed in lines 296-299 in the text. It is also interesting to study
an adaptive scheduling strategy of $\beta$, and we leave it as a future work.

[Meta-Review · NeurIPS 2020]

The reviewers appreciate the simple idea brought up in the paper and the experiments designed to understand its effect and the theoretical justification. Some reviewers did express concerns regarding the significance of the theoretical results and the concerns remain after the rebuttal. Please try to incorporate these feedback in your final draft.